# On the Supposed Mass of Entropy and That of Information

**DOI:** 10.3390/e26040337

**Published:** 2024-04-15

**Authors:** Didier Lairez

**Affiliations:** Laboratoire des Solides Irradiés, École Polytechnique, CEA, CNRS, IPP, 91128 Palaiseau, France; didier.lairez@polytechnique.edu

**Keywords:** thermodynamics, information theory, Landauer’s principle

## Abstract

In the theory of special relativity, energy can be found in two forms: kinetic energy and rest mass. The potential energy of a body is actually stored in the form of rest mass, the interaction energy too, but temperature is not. Information acquired about a dynamical system can be potentially used to extract useful work from it. Hence, the “mass–energy–information equivalence principle” that has been recently proposed. In this paper, it is first recalled that for a thermodynamic system made of non-interacting entities at constant temperature, the internal energy is also constant. So, the energy involved in a variation in entropy (TΔS) differs from a change in the potential energy stored or released and cannot be associated to a corresponding variation in mass of the system, even if it is expressed in terms of the quantity of information. This debate gives us the opportunity to deepen the notion of entropy seen as a quantity of information, to highlight the difference between logical irreversibility (a state-dependent property) and thermodynamical irreversibility (a path-dependent property), and to return to the nature of the link between energy and information that is dynamical.

## 1. Introduction

The link between information and energy finds its origin in Maxwell’s demon, who, by acquiring information about a thermodynamic system, is able to act on it and produce work in return [1]. Later, Shannon [2] formalized this link mathematically considering the quite different problem of information processing. He demonstrated that the minimum average number *H* of a bit to encode a random variable emitted by a source, let us say the current microstate of a dynamical system, is equal to a factor ln2 to the Gibbs entropy *S*, that is itself equal to the Clausius entropy of the system: S=Hln2 (in this paper, *S* is dimensionless and temperature *T* is in Joule). Hence, the link: from the second law of thermodynamics, acquiring one bit of information about a dynamical system has a minimum energy cost equal to Tln2 that can, in return, be potentially used to extract, at best, the same quantity of energy from the system.

Landauer followed by Bennett [3,4,5] tackled the problem in a quite different way. In their approach, the logical states 0 or 1 of one bit of information correspond necessarily to two different thermodynamic states. Moreover, any irreversible logical operation, such as erasing one bit, corresponds to an irreversible non-quasistatic thermodynamic process that consequently has a non-zero minimum energy cost when performed cyclically. This is the so called “Landauer principle”. In this way, it is believed that “*Information is physical*” [6] in a more convincing manner than with Shannon’s information theory.

Based on the Landauer–Bennett idea, a new step (in the wrong direction) has recently been taken. Information stored in the form of physical bits is considered a kind of potential energy to which, in the framework of special relativity, it can be assigned a mass [7,8,9,10]. This is the “mass–energy–information equivalence principle” that “*states that information is a form of matter, is physical, and can be identified by a specific mass per bit while it stores information. It is shown that the mass of a bit of information at room temperature (300 K) is $3.19\times 10^{-38}$ Kg.*” [7]. This idea has been criticized [11] at an epistemological and ontological level: what exactly does “physical” mean in “*Information is physical*” ? [6]. The aim of this paper is to show that this idea is also false for at least three reasons, which this time are at a more prosaic level and are developed in the three sections of this article.

In the first section, the “mass–information equivalence principle”, is addressed from the thermodynamic side as a “mass–entropy equivalence principle”. It recalls the basic difference between potential energy and entropy: the elastic energy of a spring is fundamentally different from that of a rubber or from that of a compressed volume of gas. For a spring, it originates from a microscopic interaction potential, whereas it is emergent for a rubber or a gas. It will be shown that a monothermal variation in entropy (TΔS) of a body is not accompanied by any variation in its mass.

In the second section, the “mass–energy–information equivalence principle” is addressed at its root, that is to say the Landauer principle. In a previous paper [12], it was shown that the Landauer–Bennet idea cannot be a general principle but is only true in a particular case. It follows that any derivative of this “principle” is logically ruled out. Here, new examples will be given to illustrate that logical and thermodynamical irreversibilities are uncoupled. In fact, as defined by Landauer himself, the logical irreversibility of an operation is intrinsic to its initial and final states and is independent of the path used to achieve the operation, in contrast to the thermodynamic irreversibility that is a property of the path.

In the third section, the last argument against the “mass–energy–information equivalence principle” is given: the link between information and energy is valid for fresh information about a dynamical system. Old information, or information detached from its subject matter, is no longer information and has no value.

## 2. Potential versus Entropic Forces

After Shannon [2], we know that the thermodynamic state quantity *S*, named entropy and introduced by Clausius [13] to account for exchanges of heat during a process, is to a factor ln2 mathematically equal to the minimum average number *H* of bits necessary to encode in which microstate the dynamical system is currently, namely, the quantity of information emitted by the system.
(1)S=Hln2
Even though Shannon’s information theory is not used by Landauer and Bennett, they do not question its correctness. It follows that the hypothetical “mass–information equivalence” is nothing other than a “mass–entropy equivalence” that can be addressed in a pure thermodynamic framework in the context of special relativity. This is the aim of this section.

In the theory of special relativity, the energy of a body takes two forms: kinetic energy and mass (or rest mass, or rest energy). Mass is the energy that is stored by the body when it is at rest. For any monothermal transformation, the product of temperature *T* (in Joule) to the variation in entropy ΔS (dimensionless) has the dimension of an energy. Behind the idea of a “mass–entropy equivalence” is that TΔS is a sort of potential energy stored (or released) somewhere in the system at the end of the transformation, to which can be attributed an equivalent mass difference $T\Delta S/c^{2}$ (where *c* is the celerity of light), by virtue of Einstein’s famous equation. Before addressing this analogy, let us first deal with the case of potential energy in mechanics.

### 2.1. Mechanical Potential Energy

When a force is applied to a body over a given distance, mechanical work is done; that is to say, energy is transferred from one body (the one applying the force) to another (the one we are interested in). But an important point is that “*Work is a process; once done, it no longer exists. It is something that cannot be stored; what is stored is energy*” [14].

When a stone is transported from the ground to a table, mechanical work is done against the force of gravity. The energy transferred to the stone is recovered in the form of kinetic energy when the stone falls back down. If energy is conserved, where is it between these two processes? We usually say that it is stored in the form of potential energy in the earth–stone system. But, as noted by Hecht [15], kinetic energy can be measured, as well as the work that has been done, without affecting their integrity, but the potential energy of the stone on the table cannot. When we measure it, it is no longer potential energy, it is kinetic energy. Potential energy is a concept that was introduced to ensure the conservation of energy, the energy is actually stored in the form of mass, a physical quantity that can, in principle, be measured without affecting its integrity. For instance, the mass equivalence of potential energy can be measured for nuclear fission—the mass of a nucleus is smaller than the sum of those of its nucleons taken independently. The difference is due to the attractive strong interactions between nucleons and is divided between the different parts when they separate. Even if it is not measurable for a stone on a table, for consistency of the theory, we are obliged to assume the same effect. The stone has more mass on the table than on the ground.

The elastic potential energy of a constrained metal spring is of the same nature (see Figure 1). “*When work is done on the spring, the spring’s rest energy increases in the form of Δm*” [15]. Compared to the gravitational potential energy of a stone that has an arbitrary zero at the ground level, the spring can be stretched or compressed with an identical restoring force (up to the sign) towards the equilibrium position that unambiguously defines the zero of the potential energy of the spring. This equilibrium position originates from the microscopic net interaction potential between the atoms of the crystal—each atom is in the minimum of a potential well made by the presence of others. The work necessary to constrain the spring is that needed to deviate atoms from this minimum. The potential energy of the spring is the sum of those of its atoms.

Not only metal springs are elastic. So are pieces of rubber. But contrary to what is suggested in ref. [15] (but this point is marginal in the paper), the origin of this elasticity is different. It has the same origin as that of a volume of gas in thermal equilibrium with its surroundings but at a different pressure. It is entropic [16].

### 2.2. Entropic Forces

Consider a closed volume of gas in a container. That is to say “a closed system” in the thermodynamic sense of the term, a system with a constant quantity of matter that can exchange energy with its surroundings. Throughout this paper, by “system”, we mean such a closed system (excluding its surroundings) and all energy balances will be signed in relation to it.

Let us equip the container with a piston allowing its contents to be compressed or expanded. Like the spring, the piston has an equilibrium position that corresponds to equal forces applied on it. When deviating it from this position by pushing or pulling, we feel an elastic restoring force that is apparently comparable to what it would be if the gas were replaced by a spring. So, it is legitimate to state that when the piston deviates from its equilibrium position, the overall system stores an amount of elastic potential energy. But at a microscopic level, for a perfect gas, there is no interaction potential between molecules. Even for real gas, for which pair-interactions can be modelized by a Lennard–Jones potential, interactions can be neglected as soon as the particles are not in contact (between two collisions). Gas particles do not interact at a distance and do not have an equilibrium position.

The elastic force we feel on the piston is due to the balance between the many collisions it experiences with the gas molecules on both sides (each collision involving the kinetic energy of one given particle) and the force applied by the surroundings on the piston.

First, note that the internal energy of a system made of *N* non-interacting independent entities is its temperature *T* (in Joule). For the sake of simplicity, let us assume that this is the case for the gas inside the container and for the atmosphere outside (that of the surroundings), which is reasonable in the case where both are air close to atmospheric pressure. Neglecting interactions means, in particular, neglecting hydrodynamic interactions, friction, and viscosity and, thus, the time delay to reach the equilibrium after any perturbation. In terms of thermodynamics, it means that the overall system (gas plus surroundings) is always at equilibrium and that the transformation is reversible. The existence of such a reversible process able to pass from one state to another is the only way in phenomenological thermodynamics to measure (and thus to define) entropy that is given by its exact differential:(2)$$\;dS=\frac{\;dQ_{r}}{T},$$
where $Q_{r}$ is the heat exchanged for a reversible process.

Note that the notion of instantaneous equilibrium, and consequently that of reversible process, appears to be incompatible with special relativity because, in principle, nothing (and, in particular, the propagation of a perturbation) can go faster than light. The same issue exists for mechanical potential energy [17,18]. But this is not a problem as far as we are concerned by the initial and final states of a process and not by the process in itself (e.g., as far as we consider monothermal and not isothermal processes). The notion of a reversible process in thermodynamics is equivalent to the classical mechanics limit of special relativity, which is conditioned on the basis of two assumptions: the velocities of particles are small compared to that of light; the characteristic distances in the system are small, as are the delays in the propagation of signals.

To go further into our problem, two cases for the gas container are worth considering:the container is adiabatic, i.e., it prevents heat exchanges with the surroundings;the container is diathermal, i.e., it allows heat exchanges with the surroundings.

Consider the gas in an adiabatic container. Compressing the gas by pushing the piston, we produce work and provide to the gas an equivalent amount of energy. Doing so, as the gas cannot dissipate heat, its internal energy necessarily increases, and so does its temperature. From Equation (Equation 2), it follows that if there is no heat exchanged, there is no variation in entropy. Clearly, the reversible adiabatic case is not associated with “mass–entropy equivalence” that envisages differences of entropy between two states at the same temperature, but it is worth considering for what follows.

Can the increase in internal energy (temperature) be assimilated to the elastic potential energy? No, it cannot. Because when pulling (instead of pushing) the piston, this time, the gas decreases in internal energy, but there is still a restoring force and a positive potential energy stored somewhere. The elastic potential energy is, in fact, stored in the solid container, and in the external mechanical part of the device that drives the piston, in a form quite comparable to that of an elastic spring, but it is not stored in the gas.

Indeed, when considering TΔS and its possible “mass–equivalence”, implicitly, one considers a system in which temperature is kept constant thanks to heat exchanges with the surroundings. Consider a closed volume of gas in an ideal diathermal container. Heat exchanges ensure that for any transformation, the initial and final states of the gas are both in thermal equilibrium with the surroundings. If in addition, the surroundings are so large that their temperature can be considered constant, the transformation is monothermal—the temperatures of the initial and final states are the same. Then, by definition, the internal energy of the gas is also unchanged. Whatever the work provided to it, the gas does not store additional energy compared to what it initially contained. The system can receive work *W*, but it dissipates an equal amount of energy to the surroundings in the form of heat *Q*, or does the reverse. For any monothermal transformation:(3)W+Q=0

The entropy change is given by considering a process slow enough to ensure a constant temperature at all times (isothermal transformation). By integrating Equation (Equation 2), one obtains
(4)$$T\Delta S=Q_{r}=-W_{r}$$
The differential of the work is $\;dW_{r}=-P\;dV$, with *P* the pressure and *V* the volume. As $P=NTV^{-1}$, one has $\;dW=-NTV^{-1}\;dV$, and integration from $V_{0}$ to *V* gives
(5)$$T\Delta S=TN\ln(V/V_{0})$$
Thus, in the monothermal case, this time under the action of the piston, the gas undergoes a change in entropy, but the corresponding elastic potential energy is still not stored by the gas itself. It is stored by the container, the mechanical part that drives the piston, and the surroundings. The gas itself does not store more or less potential (or internal) energy, which could correspond to any variation in its mass. In this, it differs from the spring (see Figure 1 and Figure 2).

The case of a piece of rubber is even more illustrative because there is no need of a container. Rubber is made of cross-linked linear polymer chains which form a three-dimensional network. Let *N* be the number of independent chain segments of the chain portion between the two first neighbor cross-links at a distance $R_{0}$ when the rubber is unconstrained. In this state, the chain conformation is random with $R_{0}$ and *N* linked by the scaling relation $N\propto R_{0}^{2}$. Stretching the rubber, causes the distance between the cross-links to increase in the same direction, forcing the chain to be less random. The corresponding variation in entropy is as follows:(6)$$\Delta S\propto -{\left(R/R_{0}\right)}^{2}$$
But, as for the gas, at constant temperature, the internal energy is constant ([16] p. 31). It follows that the elastic potential energy, even if it originates from the rubber, is not stored inside the rubber but by the mechanical part of the surroundings that is responsible for its stretching.

This result for a gas or a piece of rubber is actually general. The internal energy of a set of independent entities, such as a set of bits, is its temperature. It follows that for a thermodynamic closed system, any monothermal variation in entropy, interpreted in terms of the potential energy stored in the form of rest mass, cannot be localized within the system itself, but only in its surroundings.

## 3. Logical versus Thermodynamical Irreversibilities

The second law of thermodynamics has two parts. The first is the definition of entropy as a state quantity defined by its exact differential given by Equation (Equation 2) valid for a reversible process. The second accounts for the general case. It is the Clausius inequality that applies to a closed system at constant temperature:(7)−Q≥−TΔS
This means that the quantity −Q of heat dissipated by the system (and received by the surroundings) is always higher than −TΔS. For a closed system made of independent entities at constant temperature, the internal energy is also constant (Equation (Equation 3)), so that the heat dissipation is compensated by the same amount of work (W=−Q) provided to the system. Generally, heat is unwanted and work is more valued and can be viewed as an energy cost. With Equation (Equation 1), the Clausius inequality is rewritten in this case as:(8)W≥−TΔHln2
Consider the process of reducing the volume of the phase space of a dynamical system (ΔS≤0). The uncertainty about the current microstate of the system, or the quantity of information it emits, is reduced by ΔS/ln2. The total amount of information we lack to describe the system is reduced accordingly, as if we had acquired ΔH=ΔS/ln2 bits of data about the system. So, Equation (Equation 8) can be expressed per bit (ΔH=−1) of acquired data:(9)$$W_{\mathrm{acq}/\mathrm{bit}}\geq T\ln2,$$
which expresses that Tln2 is the minimum energy cost to acquire 1 bit of data about the dynamical system under consideration. This statement is nothing more than a reformulation of the Clausius inequality in terms of the quantity of information. Actually, there is absolutely no difference between the Clausius entropy and the Shannon quantity of information emitted by the system (the source) about its current microstate.

Equation (Equation 9) suffices in itself to account for the functioning of demonic engines like that of Maxwell [1] or the simplified version of Szilard [19] (see Figure 3) and their physical implementations in the form of ratchet–pawl mechanisms [20,21]. Each bit of information needed for the engine to work has a minimum energy cost of Tln2. This is a direct application of the second law that provides the link between information and energy.

The Landauer principle [3,4,5] reaches the same result without information theory but with an indirect application of the second law. It is at the heart of the “mass–energy–information equivalence principle” [7,9]. Let us report some quotes:

*The M/E/I principle [the mass–energy–information equivalence principle] states that information is a form of matter, it is physical, and it can be identified by a specific mass per bit while it stores information or by an energy dissipation following the irreversible information erasure operation, as dictated by the Landauer principle.* (Vopson [9])

*We demonstrated that the Landauer principle supplies the estimation of the minimal mass of the particle allowing the recording/erasure of information within the surrounding medium at temperature T.* (Bormashenko [22]).


The purpose of this section is not to discuss the reasoning leading to these conclusions, but rather to show that the root of them, i.e., the Landauer principle, is not correct because it results (1) from considering a doubly particular case, and (2) from a confusion between logical and thermodynamical irreversibilities.

### 3.1. Landauer “Principle” Is a Doubly Particular Case

The Landauer principle, allowing Equation (Equation 9) to be found without any reference to the Shannon’s encoding problem, is based on the two assumptions below:For a cyclic process (such as that of the Szilard engine), the recording or acquisition of a data bit is supposed to first require the erasure of that bit.The erasure of a data bit is supposed to necessarily involve an irreversible non-quasistatic stage (i.e., uncontrollable and similar to the free expansion of a gas), so that when performed cyclically, it has a minimum energy cost of Tln2.

The first supposed requirement will be discussed in the second part of this section. Here, we only focus on the second.

Landauer and Bennett first imagine a one-to-one correspondence between logic and thermodynamic states. They imagine a particle in a bi-stable potential well allowing two different positions (labeled 0 and 1, respectively) to be equally stable. The ERASE operation consists in putting the particle in position 0 (SET TO 0). It is performed in three elementary stages or operations.

Landauer’s ERASE procedure:SET TO S (standard state): set the particle in an undetermined position by lowering the energy barrier between the two positions.BIAS TO 0: apply a bias favoring the zero position.STABILIZE: raise the energy barrier and stop the bias.

The point is that the path chosen by Landauer to achieve the first stage throws the probability density of the particle position out of control and causes it to leak from one potential well to the other. It is similar to the free expansion of a gas, initially confined in one half-volume of a box (with label 0 or 1) and suddenly allowed to occupy the entire volume. The last two stages can be performed in a quasistatic manner equivalent to the isothermal compression of a closed volume of gas in the correct half-volume of its container (with label 0). During the first stage, neither heat nor work are exchanged with the surroundings, whereas the last two have an energy cost at least equal to Tln2. The net energy balance of the total operation is thus:(10)$$W_{\mathrm{erase}}\;1\;\mathrm{bit}\geq T\ln2$$
Conjointly with the necessity to erase prior to acquiring 1 bit of data, this last equation allows us to derive Equation (Equation 9).

Here, let us first outline a common error. Equation (Equation 10) (Landauer’s principle) and Equation (Equation 9) (that directly comes from the Clausius inequality Equation (Equation 7)) both apply to a thermodynamic closed system and appear strongly alike, so that the Clausius inequality and Landauer’s principle are often confused (see e.g., [23]). Consider a cycle as Landauer explicitly does ([3] Section 3). Thus, ΔS=0 and −Q≥0, according to the Clausius inequality. But following Landauer, if the cycle involves the erasing of a bit, the heat dissipated is always −Q≥Tln(2). This finite limit (known as Landauer’s limit) comes from the non-quasistatic irreversibility of the first stage of Landauer’s ERASE procedure that is supposed by Landauer and Bennett to be unavoidable. It is precisely what is disputed in this section.

In a previous paper [12], it was shown that the best way to avoid any probability-density leakage between the two potential-wells is to have only one, but still two logical states. This is subject to the possibility of establishing a two-to-one correspondence between logic and thermodynamic states. An example of such a correspondence has been given in ref. [12], which undermines the generality of the Landauer principle. For the purpose of this paper, let us give another counter-example.

In Figure 4, we imagine a cam that can compress two springs (which can be replaced if we want by two volumes of gas compressed by pistons). The angle of the cam defines the bit state, whereas the state of the springs defines the thermodynamic (or mechanical) state. The cam has a smooth shape that continuously passes from an elliptical section on one side (front) to a circular one on the other (rear), both being centered on the axis of rotation provided with a steering wheel allowing it to be driven. When the steering wheel is pushed (left and right in Figure 4), the bit is stabilized in position 0 or 1 by the two springs. In both positions, the constraints they undergo are the same and it is not possible to know the bit-state by simply observing the state of the springs. There is actually only one thermodynamic state and, thus, a two-to-one correspondence between logic and thermodynamic states. The ERASE operation can be performed by pulling the steering wheel, so that the springs are in contact with the circular section of the cam and the energy barrier between the two logical states is zero; the bit is then in the undetermined S-state. Then, the steering wheel is turned in order to align the red mark of the cam with position 0 (apply a bias); finally, the steering wheel is pushed to restore the energy barrier. The entire operation only involves friction, so that the heat dissipation tends to zero as the velocity of the steering wheel manipulation tends to zero. It is quasistatic. Note that if in addition, the small radius of the ellipse is equal to the radius of the circle (as in Figure 4), the state of constraint of the springs is the same whether the bit is in position 0, D, or 1. So that the entire ERASE operation can be performed at constant elastic potential energy of the springs. Note also that in the two logic states (0 and 1), as the elastic potential energies of the springs are the same, their mass in the framework of special relativity is also the same. A set of independent bits, built with this physical implementation, can be erased in a quasistatic manner and with no variation in the rest mass.

A two-to-one correspondence could be viewed as a very particular case of physical implementation of logic and the one-to-one correspondence perceived as the general case. Interestingly, the example just given above can be slightly modified in order to obtain a one-to-one correspondence similar to that of Landauer but with no leak. For this, it is enough to decenter the elliptical face of the cam (while the circular face remains centered), as shown in Figure 5. Then, the logical states 0 or 1 are still stable and well separated by an energy barrier, but the state of the springs is not the same in both cases. It is now possible to know the bit-state by only observing in which state the springs are (compressed up or down). There is a one-to-one correspondence between the logic and thermodynamic states. The procedure to change the bit state (e.g., SET TO 0 or ERASE) is the same as in the previous case and is quasistatic. During this operation, the potential energy of the springs is first released (when pulling the steering wheel), then the same amount is stored again (when pushing the steering wheel). The sum of the potential energy of the springs is the same whatever the bit state so that it can be set to 0 with no change in rest mass.

From these two counter-examples, it appears clearly that logical and thermodynamical irreversibilities are not linked. The bit can be irreversibly erased in a quasistatic manner that tends to be thermodynamically reversible if performed slowly enough. This result is not a sleight of hand, as it could be, for example, by changing in a hidden way what we mean by “system”—for instance, by considering a (sub)system for the logical process and the whole system (including the surroundings) for the thermodynamical process. This result originates from a fundamental difference between logical and thermodynamical irreversibilities, as explained below.

### 3.2. Irreversibilities

The logical irreversibility of an operation is defined by Landauer: “*We shall call a device [an operation] logically irreversible if the output of a device does not uniquely define the inputs.*” (Landauer [3]). The ERASE operation is logically irreversible: two possible initial states 0 or 1 (the input of the operation) lead to only one final state 0 (the output). Further, in the same paper, Landauer writes: “*Logical irreversibility, we believe, in turn, implies physical [thermodynamical] irreversibility*”. This last point is discussed in this section.

As soon as we deal with the physical implementation of a logical operation on a bit, this operation becomes a thermodynamical transformation (or a process). With the above definition, it is clear that the logical irreversibility of an operation is defined only by the properties of the bit before and after the transformation. The logical irreversibility is a property of the initial and final states of the bit. It is not a property of the path that has been used to perform the transformation. In other words, a transformation (say a transport) from *A* to *B* is logically irreversible if once in *B*, the information from where the system started has been lost, so that it is not possible to return to the starting point *A*. The same transformation is thermodynamically irreversible if it is not possible to return to *A* by using the same path backward. Thermodynamical irreversibility is a path property.

Due to this fundamental difference, logical and thermodynamical irreversibilities cannot be linked by a material implication.

In thermodynamics, when we deal with the path allowing a system to change from an initial to a final state, we first wonder whether or not the path can be decomposed in a succession of infinitely small changes, i.e., in a succession of quasi-equilibrium states. In other words, considering the variation in entropy versus the extent of the change, we wonder whether or not this variation is differentiable. If so, the path is said to be quasistatic and can be potentially reversible if slowed down enough. Otherwise, the very point where the discontinuity occurs is an inherently irreversible step. At this point, the process evolves spontaneously in an uncontrollable manner. This occurs when a system suddenly finds itself far from equilibrium when an internal constraint has been released (a typical example is that of the free expansion of a gas). Consider such an irreducible step as a process in itself. No work can be extracted from it (W=0). At a given temperature, according to Equations (Equation 3) and (Equation 7), one obtains:(11)ΔS>0
An increase in entropy is, in fact, a necessary condition for an inherently irreversible (non-quasistatic and irreducible) process to occur. But it is not at all a sufficient condition. The same change in entropy can occur using a reversible path, otherwise it would not be defined in thermodynamics because it could not be associated with any measurable quantity.

Consider the ERASE operation on the thermostatistics side and let Γ be the phase space. The two initial possibilities Γ={0,1} result in only one Γ={0} once the bit has been erased. So that
(12)$$\Delta S_{\mathrm{ERASE}}<0$$
This means that ERASE is not inherently irreversible, but that any thermodynamic path leading to this operation can be decomposed into elementary steps (just as Landauer did). In fact, in Landauer’s physical implementation, only the first step (the one which brings the system into an indeterminate state S) can possibly be intrinsically irreversible, because Γ initially {0,1} changes into [0,1], so that
(13)$$\Delta S_{\mathrm{SET}\;\mathrm{TO}\;D}>0$$
For this step, another path than that of Landauer can be chosen, quasistatic this time, as shown in Section 3.1.

“*[Logical irreversible] operations are quite numerous in computer programs as ordinarily written: besides erasure, they include overwriting of data by other data*” (Bennett [4]). On the thermostatics side, the status of OVERWRITE depends on which new data replace the old ones. For cyclic recording of data, such as that needed when implementing a Szilard engine, if the system is in a stationary regime, the probability distribution of the data is unchanged from one cycle to the next. So that overwriting old data by new ones leaves the phase space unchanged. Thus,
(14)$$\Delta S_{\mathrm{OVERWRITE}}=0$$
It follows that OVERWRITE is also not an inherently irreversible process. It can be decomposed in quasistatic steps. To “elucidate” the functioning of the Szilard engine, Landauer–Bennett chose to break down OVERWRITE into ERASE then WRITE, which brings us back to the previous discussion about ERASE. But OVERWRITE can be performed in a direct manner without ERASE, just as an old magnetic tape can be reused without needing to be reset in a virgin state (erased).

Logical irreversibility is not linked to thermodynamic irreversibility. There is no conceptual impediment for a logical irreversible operation to be quasistatic and for heat dissipation to vanish as the operation slows down.

Let us conclude this section with some quotes from the recent literature about Landauer’s principle: “*Experimental verification of Landauer’s principle linking information and thermodynamics.*” (Bérut et al. [24]). “*Information and thermodynamics: experimental verification of Landauer’s Erasure principle.*” (Bérut et al. [25]). “*We experimentally demonstrate a quantum version of the Landauer principle.*” (Yan et al. [26]). “*Landauer’s principle has been recently verified experimentally*” (Binder [27]). Clearly, the problem of induction in natural sciences [28,29] seems to be forgotten. The results of a peculiar experiment either agree with a theoretical statement—in this case, it confirms this statement but cannot prove its generality—or disagree with this statement—in which case it refutes the statement and proves that it is not general. Actually, on this point, “true” and “false” are not symmetrical, just like “existence” and “non-existence”—you can prove that something exists (by finding it), but you cannot prove that it does not exist. Logical irreversibility is a property of the initial and final states of a process, whereas thermodynamical irreversibility is a path property. It follows that for a given logical irreversible process (i.e., two given states), you can find an irreversible thermodynamic path (e.g., the experiments that confirm the Landauer principle), but this does not mean that a reversible path does not exist (examples of such paths were given in this section).

*Parenthesis:* One might ask why not consider in this section a completely isolated bit (or set of bits) rather than the more complicated situation of a bit plus its surroundings? Indeed, it is not possible to envisage an irreversible logical operation on a completely isolated bit (or set of bits). Isolated systems only undergo spontaneous transformations that occur without external intervention. But the notion of a bit-state (inherent in that of information storage) assumes stability and not spontaneous evolution towards another state.

## 4. Information Is Dynamical

“*To test the hypothesis [the mass–energy–information equivalence principle], we propose here an experiment, predicting that the mass of a data storage device would increase by a small amount when it is full of digital information relative to its mass in the erased state. For a 1 Tb device, the estimated mass change is $2.5\times 10^{-25}$ Kg.*” (Vopson [7]). Beyond the already recognized difficulty of carrying out such a measurement [9], we will show here that this idea is nonsense and inconsistent with everything we know about thermodynamics (and physics).

The first argument directly comes from the fundamental difference between logical and thermodynamical irreversibilities that has been exposed in the previous section. Consider the above hard drive full of data and the three experiments below:Directly erase the hard drive. This operation is logical irreversible (it is impossible to retrieve the data once they have been erased).Make a copy of the hard drive then erase the original one. This operation is logical reversible (with the copy, it is possible to restore the original hard drive to its original state).Make a copy, erase one hard drive then the second. The erasure of the first hard drive is logical reversible, whereas that of the second is irreversible.

For the copy to be of any use in preserving data integrity, it must be physically independent from the original (we can imagine moving it to the other side of the Earth). It follows that the mass defect (if there is one) that would be measured for the above four erase operations would have exactly the same value. If there is a mass defect, it has nothing to do with logical irreversibility nor with information that would be lost or not.

The independence of two hard drives also holds for two bits. This is implicit in the mass–energy–information “equivalence” when it is expressed per bit: “*Using the mass–energy–information equivalence principle, the rest mass of a digital bit of information at room temperature is $m_{bit}=3.19\times 10^{-38}$ Kg.*” (Vopson [8]). But sometime it is explicit: “*Essentially, a bit of information could be seen as an abstract information particle, with no charge, no spin, and rest mass*” (Vopson [8]). It follows that the above three experiments performed with a hard drive could be performed with a bit with the same conclusion.

If data are stored with bits set either to 0 or 1, the two values equally participate in the storage of information. If a bit of information has a rest mass, the latter is independent of its value 0 or 1. The ERASE operation is usually presented as a SET TO 0 operation, but this is a convention and it could be SET TO 1 (as Landauer did in his first paper [3]). This suggests another experiment in two stages:Erase (SET TO 0) one given bit of information (with value either 0 or 1).Erase it a second time.

According to the mass–energy–information equivalence principle, a mass defect should be observed in the first stage, while it should not be observed in the second. The only explanation for this difference should be the change in entropy caused by the ERASE: in the first stage Γ:{0,1}→{0}, whereas for the second stage Γ:{0}→{0}. This brings us back to the first section of this paper: at constant temperature no change in rest mass accompanies a change in entropy.

In fact, the data stored in the hard drives or the bits above are not information in the sense given to that word by Shannon. The physical support of these data can be considered as a thermodynamic system in its own right. But for this, it must be read and emit outcomes just like other dynamical systems do. To consider these data as information, they must not be detached from their subject matter, i.e., from the dynamical system that emits this information. Once detached from this dynamical system, the information becomes frozen and outdated, it has no value and no link with energy. Let us emphasise this point.

In the word “thermodynamic”, there is “dynamic”. This is a truism, but apparently worth remembering: ΔS must be multiplied by temperature *T* to become an energy. In other words, the link between energy and information only exists when the renewal of the latter obeys the same dynamics as that of the system it concerns. This is particularly clear with the Szilard engine. Imagine that the position of the particle is recorded on a hard drive for a given time interval. Once this is done, these old data cannot be of any utility to extract energy from the current system in return to that spent on their acquisition and recording.

## 5. Conclusions

Entropy (and quantity of information) is a concept. Just like potential energy is. There are actually many common points between them. For instance, just like potential energy, it is not possible to measure entropy without changing it into something else (i.e., changing potential energy into kinetic energy and changing entropy into heat). Also, zeros for both quantities may appear arbitrary and not intrinsic to the nature of things. Nevertheless, these concepts have some fundamental differences. Potential energy was introduced to satisfy a conservation principle for energy (first law of thermodynamics), while entropy was introduced to account for the irreversibility of changes in the form of energy (second law), that is to say, a change in quality but not in quantity. Basically, this difference rules out the idea of a mass–entropy equivalence (Section 2).

This idea of a mass–entropy equivalence (or mass–information equivalence) is actually the last attempt to materialize the link between information and energy, that is to say, to make it more “physical”, more tangible, less elusive. It originates from the Landauer principle. The latter is actually due to a confusion between logical irreversibility (that is a state-dependent property) and thermodynamical irreversibility (that is a path-dependent property). Once this has been clarified, it appears that there is no finite limit of heat dissipation to erase a bit. In other words, a bit does not store more energy whether it is set to a given data value or erased (Section 3).

This brings us to the last confusion at the origin of the mass–information equivalence—stored data is not information in the Shannon sense. Stored data are frozen, information is dynamical. Stored data are actually outcomes of a dynamical system that have been acquired (thus, brought to our knowledge) then copied somewhere (stored). But information is very special, as soon it has been given (acquired), it is no longer information, it is outdated. Information cannot be given twice. The link between energy and information is that of a dynamical system as a source of information in the spirit of Shannon (Section 4). Just like the link between energy and entropy.

In 1948, when Shannon [2] identified the minimum number of bits (which he called the quantity of information) to encode the behavior of a dynamic system as its statistical entropy, this was a great advance—entropy became information. Although this alone was of great importance for computer scientists or for communications engineers, for physicists, the major breakthrough did not lie in this identification, which may appear to them as a simple question of vocabulary. The major breakthrough was in the second part of the work of Shannon, who also identified this quantity of information with a measure of the uncertainty about the system. The resulting principle of maximum entropy [30,31,32] made it possible to legitimize a priori probabilities and resolve many inconsistencies in statistical mechanics (for a review on this topic, see [33]).

Entropy is information. As fascinating as that may be, we must not forget that the relationship also holds in the other direction—information is entropy and is just that. 

## Figures and Tables

**Figure 1 entropy-26-00337-f001:**
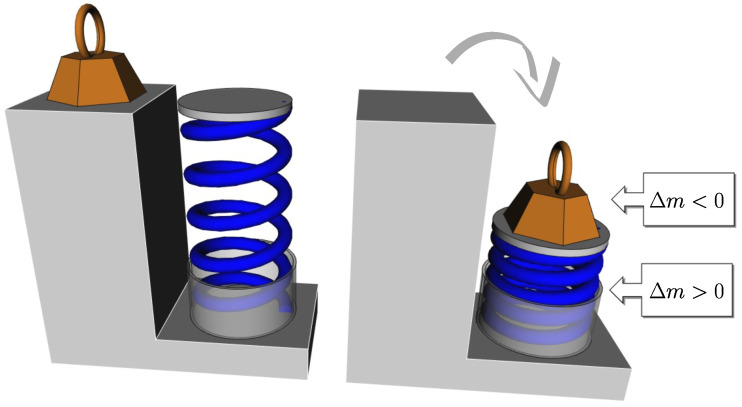
When a weight (in brown) compresses a vertical spring (in blue), it undergoes a decrease in mass (less potential energy), while that of the spring increases (more elastic potential energy).

**Figure 2 entropy-26-00337-f002:**
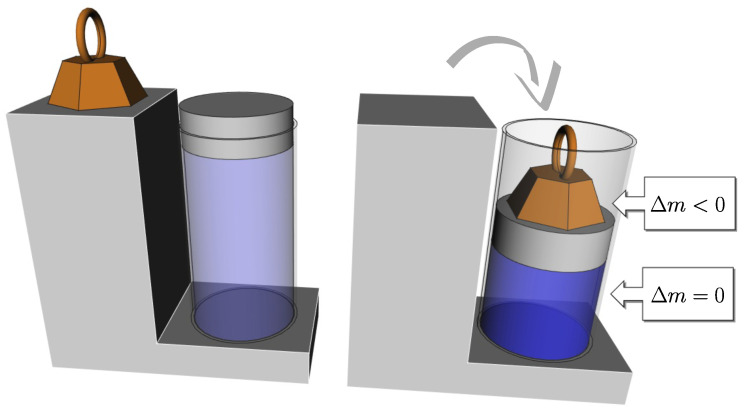
If the spring of Figure 1 is replaced by a gas compressed with a piston at constant temperature, the gas has less entropy but the same internal energy and its mass is unchanged.

**Figure 3 entropy-26-00337-f003:**
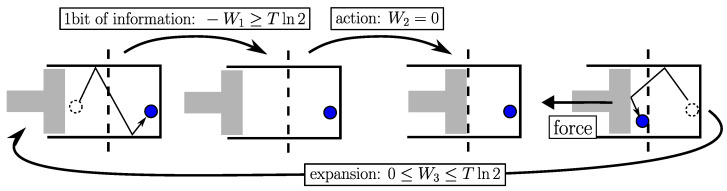
Szilard demon detects when the particle (in blue) is in the suitable compartment. In doing so, it acquires 1 bit of information for an energy cost $-W_{1}\geq T\ln2$ (Equation (Equation 9)). Then, it can install a piston for free ($W_{2}=0$), allowing the device to subsequently produce work ($W_{3}\leq T\ln2$). The overall energy balance is $W_{3}+W_{1}+W_{2}\leq 0$.

**Figure 4 entropy-26-00337-f004:**
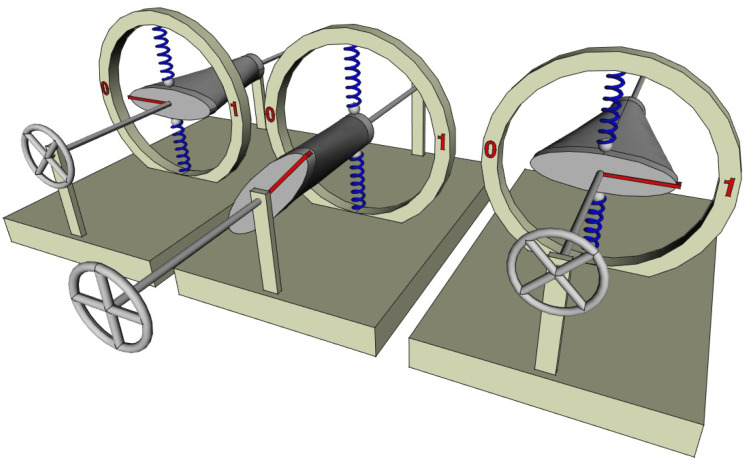
Two-to-one correspondence between logic and thermodynamic states. The bit-state (in red) is represented by the angle of a cam centered on a rotating axis controlled by a steering wheel. The cam-profile is elliptical in the front and circular in the back and can constrain two vertical springs (in blue) that define the thermodynamic (or mechanical) state. When the steering wheel is pushed (left and right), the bit has two stable logical states 0 or 1. By slowly pulling the steering wheel, the energy barrier between the two states vanishes (the potential is flat) in a quasistatic manner and drives the bit (middle) in an undetermined S-state (following Landauer–Bennet terminology). In this S-state, for a system at a molecular scale where temperature plays a role, the probability distribution of the cam angle is uniform.

**Figure 5 entropy-26-00337-f005:**
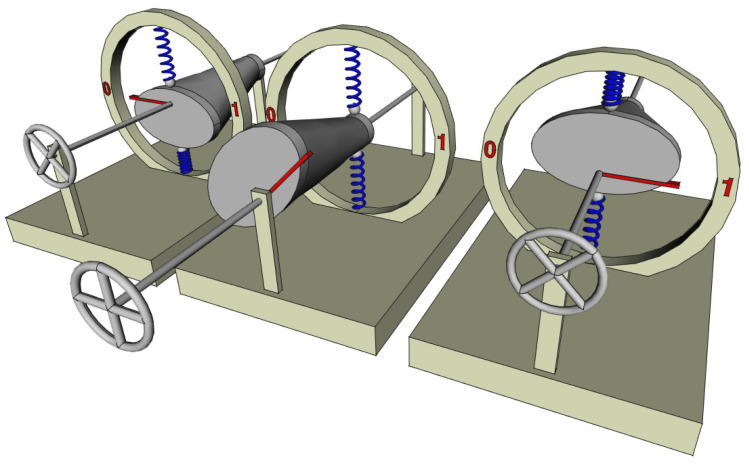
One-to-one correspondence between logic and thermodynamic states. The mechanism is similar to that of Figure 4, but this time, the cam is not centered in its elliptical part (but is still centered in its circular part). So that depending on the bit state (in red), the two springs are not equally constrained allowing to identify two different thermodynamic (or mechanical) states (in blue), each one corresponding to a different logical state. Just like for the two-to-one implementation (Figure 4), the bit can be set to 0 (erased) in a quasistatic manner that only involves friction and avoids any leakage between the two states.

## Data Availability

No new data were created or analyzed in this study. Data sharing is not applicable to this article.

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
