# Peer review of "On the Supposed Mass of Entropy and That of Information"

_entropy, 2024, doi:10.3390/e26040337_

Round 1

Reviewer 1 Report

Comments and Suggestions for Authors

The report can be found in the attachment.

Author Response

Please find in the attached document my detailed answer to your report.

Reviewer 2 Report

Comments and Suggestions for Authors

The paper is reasonably well presented and stimulates thinking about fundamental issues such as entropy and information. I do not think, however, that the perspective selected by the author is clearly defined (see below), I wish to hear the author’s explanations before making a clear recommendation.

The suggested approach lacks a clear definition of thermodynamic conditions creating some ambiguity in interpretations. I ask the author to distinguish logical and thermodynamic irreversibilities for a clearly defined object: let us consider a system (say several bits) within an environment. The environment and the system jointly form a supersystem that is completely isolated. According to the submitted paper, irreversibility (entropy increase) in the supersystem should be called thermodynamic, while loss of information in the system is called logical irreversibility.

The difference between these irreversibilities is related to clearly enforcing the boundaries in one case (supersystem) and not enforcing them in the other case (system). The supersystem can evolve unitarily (for the sake of clarity and simplicity, I use quantum terminology) and preserve entropy while different apparent effects can be observed in the system when the system is considered autonomously, ignoring the environment. Objective distinction between logical and thermodynamic irreversibilities needs defining these quantities for the same well-defined object (supersystem), but I do not see how this can be done. Hence, the difference between logical and thermodynamic irreversibilities is merely subjective and selective (i.e. based on focusing on some bits and ignoring the other bits).

Author Response

(The authors gave the same response as above.)

Round 2

Reviewer 1 Report

Comments and Suggestions for Authors

The revised version clearly addressed my concerns, now I recommend its publication.

Author Response

Thank you for the report.

Reviewer 2 Report

Comments and Suggestions for Authors

I appreciate comments provided by the author -- I understand the difference between the definitions given and based on either states or a path. Yet my key question has not been answered (maybe it was not clear in my first report). Please clarify the following issue.

Please apply your concepts of logical irreversibility and thermodynamic irreversibility to a completely isolated system (i.e. not interacting with its surrounding). Does logical irreversibility necessarily imply thermodynamic irreversibility in this system (then these concepts are intrinsically connected)? If it does not, please give an example of a completely isolated system with logical irreversibility but thermodynamically reversible.

Round 3

Reviewer 2 Report

Comments and Suggestions for Authors

Your explanation  in the text is perfectly fine. I recommend publication in the present form.    

Some thoughts:  It seems you are introducing an observer into the system --- logical bits of information are real when there is someone to detect or use them. This can easily become a philosophical issue --- e.g. does the moon exist when we do not see it?